# Approaches to the Development of Advanced Alloys Based on Refractory Metals

**Igor Razumovskii** [1] , **Boris Bokstein** [2] **and Mikhail Razumovsky** [1,2,*]

1    Joint Stock Company "Kompozit", Pionerskaya Street 4, 141070 Korolev, Russia
2    Department of Physical Chemistry, National University of Science and Technology (MISIS),
     Leninsky Prospect 4, 119049 Moscow, Russia
*    Correspondence: razmikhail@gmail.com; Tel.: +7-926-606-34-63

**Abstract:** The most promising directions of the development of heat-resistant alloys (HRAs) based on refractory metals are analyzed. The microstructures characteristic of HRAs, which it is advisable to form in promising alloys, are considered. The stability factors of the microstructure with respect to the diffusion coarsening of the hardening phases are discussed. Two groups of alloys are considered as the most promising HRAs based on refractory metals. First, the principles for design of HRAs based on (Pt, Ir)-Sc with heterophase $\gamma$-$\gamma'$ microstructure, where $\gamma$-matrix is a (Pt, Ir) solid solution with a FCC lattice, and $\gamma'$ is a strengthening phase with the structure $L1_2$ by analogy with Ni-base superalloys, are developed. The resistance of $\gamma$-$\gamma'$ microstructure in Ni, Pt and Ir alloys against the process of diffusion-limited coarsening is analyzed. It is shown that the diffusion permeability of Pt is several times less than that of Ni, so one should expect that Pt-based HRAs will not be inferior to Ni-based HRAs in terms of structural stability. The second group includes HRAs based on many not noble refractory metals. It is shown that solid solutions of the system (Ti, Zr, Hf, Ta, Nb) with a BCC lattice can be considered as a matrix of advanced refractory HRAs. The results of experimental studies of alloys based on (Ti, Zr, Hf, Ta, Nb) additionally alloyed with elements contributing to the formation of strengthening intermetallic and silicide phases are discussed. The issues of segregation of alloying elements at the grain boundaries of refractory alloys and the effect of segregation on the cohesive strength of the boundaries are considered.

**Keywords:** high-temperature alloys; microstructure; diffusion; structural stability; refractory metals; Pt-Ir-Sc alloys; alloys based on many refractory metals; cohesive strength

## 1. Introduction

Critical parts of gas turbine engines of aircraft are made mainly of Ni-based heat-resistant alloys (HRAs), the operating temperatures of which are fundamentally limited from above by the melting point $T_m$ located near 1450 °C [1–4]. The development of aerospace technology requires an increase in engine power, which is possible only by increasing the gas temperature at the turbine inlet. For operation in such conditions, materials with higher temperature parameters typical for refractory metals and compounds are required. When choosing the composition of new alloys, it should be assumed that their operating temperatures usually do not exceed ~0.75 $T_m$. Thus, to ensure the operability of gas turbine engines at 1500 °C, refractory HRAs with $T_m \geq 2000$ °C are necessary. The search of promising HRAs with high melting points for practical use compared to Ni-based HRAs is an urgent task of physical materials science.

Traditional HRAs based on one of the refractory metals, among which alloys based on W, Ta, Mo, and Nb [5] have the greatest practical application, have high values of $T_m$, and satisfy the criterion of $T_m \geq 2000$ °C; however, they have a number of significant disadvantages characteristic of metals with a BCC lattice. These include a low ductility at room temperature, especially characteristic of Mo and W, and low oxidation resistance.

Currently, the possibility of creating metallic HRAs with high melting temperatures based on many refractory metals (so-called "high-entropy alloys") is being actively investigated [6]. The matrix of such alloys is a solid solution with a BCC lattice, which is characterized by a tendency to brittle fracture. Therefore, it is necessary to develop approaches to the choice of alloying system for such alloys with an acceptable deformation ability.

Among HRAs based on a single refractory metal, a special place is occupied by noble metal alloys based on Pt ($T_m \approx 1769\ °C$) and Ir ($T_m \approx 2447\ °C$) with a FCC lattice, free from the disadvantages of BCC structures. In the field of structural materials, the main application of Pt and Ir is their use in coatings that protect the blades of gas turbine engines from the effects of aggressive environment at elevated temperatures under loads [7]. As industrial structural materials, Pt, Ir and their alloys are used for the manufacture of heat-resistant vessels for various purposes: for melting glass, which is used in the production of fiberglass composite materials, growing special-purpose crystals, etc. [8,9].

The structure of traditional HRAs based on Pt and Ir is a solid solution based on these metals. Such alloys have a single-phase FCC structure, and they are strengthened by a solid-solution hardening mechanism [8]. It should be noted that the early nickel based HRA Nimonic has a similar structure, in which the Ni-Cr solid solution was alloyed with refractory metals at the maximum purification from low-melting components [2,4].

However, it was later shown that higher heat resistance of metal alloys at elevated temperatures is provided by a polyphase structure [10] in which the solid solution is a metal matrix that is strengthened with intermetallic, carbide or silicide phases. It is possible to distinguish two main groups of polyphase structures characteristic of HRAs obtained using traditional metallurgical technologies.

The first group includes a structure consisting of a solid solution (matrix) reinforced with isolated particles of the second phase, which are formed during the decomposition of a supersaturated solid solution. Such a structure is formed in traditional dispersion-strengthened Ni-based HRAs. The second group includes eutectics, which are formed during the crystallization of the melt and are a "mechanical" mixture of two phases [11]. One of the phases in eutectic is usually a metal solid solution, and the second phase, which strengthens the solid solution, is various chemical compounds.

In this paper, the most promising approaches to the development of the newest HRAs based on refractory metals are analyzed. The features of the microstructure of HRAs, which provide high long-term strength of alloys, and the mechanisms of diffusion coarsening of different types of structures are considered. The design principles of refractory HRAs based on precious metals and high-entropy alloys are proposed.

## 2. Features of the Polyphase (Heterophase) Structure in the HRAs

### 2.1. Dispersion-Strengthened Alloys

The basis for choosing the chemical composition of dispersion-strengthened Ni-based HRAs is the phase diagram of binary Ni–Al alloys, as in Figure 1 [12].

In this system, there is a wide concentration range in which the formation of a two-phase $\gamma$-$\gamma'$ microstructure occurs, in which the $\gamma'$-phase is formed because of the decomposition of a supersaturated $\gamma$-solid solution during cooling. The $\gamma$-matrix is a solid solution of Al in Ni with a FCC lattice, and the strengthening $\gamma'$-phase is an ordered intermetallic compound $Ni_3Al$ with an $L1_2$ structure. The result is a coherent $\gamma$-$\gamma'$ microstructure in which isolated dispersed particles of the $\gamma'$ phase are arranged in a regular manner in a continuous $\gamma$-matrix, as in Figure 2 [13].

The most important characteristic of the ($\gamma$-$\gamma'$) structure is high structural stability—the ability of the alloy to maintain a given microstructure for a long time under the influence of high temperatures and loads. The ($\gamma$-$\gamma'$) microstructure has this property for several reasons. Firstly, both the FCC lattice of the $\gamma$–matrix and the ordered $L1_2$ $\gamma'$-phase of Ni-based HRAs do not change their crystal structure up to the melting temperature of the alloy (the $\gamma$-phase does not undergo polymorphic transformations and, in addition, the long-range order is preserved in the $\gamma'$-phase). Secondly, the coherent ($\gamma$-$\gamma'$) microstructure

of nickel HRAs, as in Figure 2, is characterized by high resistance to the process of its diffusion coarsening at elevated temperatures due to a unique and favorable combination of thermodynamic and kinetic factors.

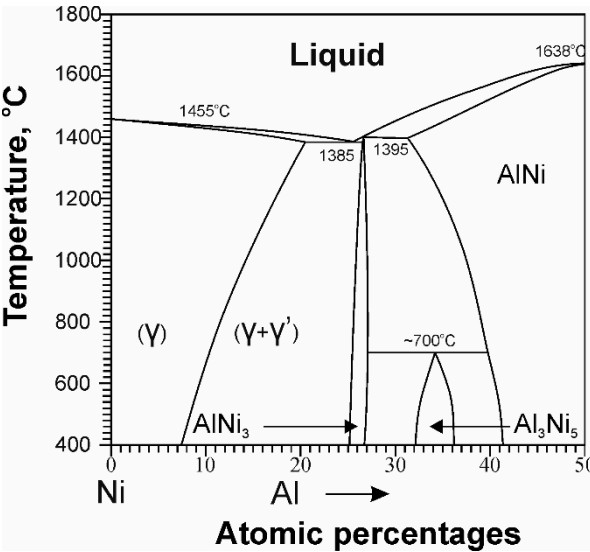

**Figure 1.** Schematic binary Ni-Al phase diagram. Taken from Ref. [12] with permission from ASM. Copyright 1986, ASM.

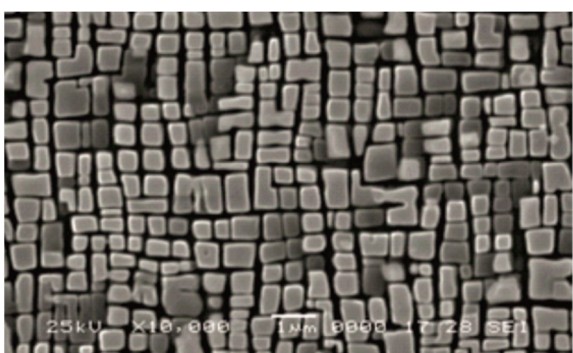

**Figure 2.** Typical coherent γ-γ′ microstructure in single crystals of Ni-16W-5Ta HRAs (mass. %); light rectangles are particles of the strengthening γ′-phase. Reprinted with permission from Ref. [13]. Copyright 2008, Elsevier.

The thermodynamic driving force of the microstructure coarsening process is the desire of the system to minimize free energy by reducing the specific area of the interfacial boundaries. In a (γ-γ′)-structure, the energy of coherent phase boundaries γ\γ′ is very low. At close values of the parameters of the crystal lattices of the γ–matrix and the strengthening γ′-phase observed experimentally in Ni-based HRAs, an "ideal" conjugation of the crystal lattices of the FCC (γ) and $L1_2$ (γ′) phases occurs at the interface; thus, the effect of misfit on the energy of the boundaries is minimized.

A favorable kinetic factor is provided by the low diffusion permeability of all elements, both the volume of the γ-matrix and the γ′-phase, and the interphase boundaries of γ\γ′ [14]. This explains the high stability of the (γ-γ′) microstructure in relation to the process of diffusion coarsening of the strengthening phase particles in the matrix.

In single crystals of Ni-based HRAs at high temperatures under the action of a uniaxial load a process of directional coarsening occurs because the particles coalesce. As a result, plate-shaped or needle-shaped (γ-γ′) structures are formed, depending on the sign of misfit or load (tension-compression). In most commercial alloys, for example, CMSX -4, the misfit is negative, and with a tensile load along the growth axis ‹100›, a plate-shaped structure

(raft) is formed [3,15,16]. The features of the process of forming a raft structure in an alloy CMSX-4 are investigated in [17], which shows that rafting in single crystals occurs even when the external load is completely removed.

### 2.2. Eutectic Alloys

The second type of heat-resistant heterophase structures is realized in eutectic HRAs, the main structural component of which is eutectic, which is a "mechanical" mixture of phases formed during solidification. One of the phases in heat-resistant eutectic is usually a metallic solid solution, which is strengthened by a second phase—silicide, intermetallic compound, and carbide.

Such a structure is formed in the Nb—Si system, in which there is a wide two-phase region (Nb) + (Nb$_5$Si$_3$) in the concentration range of Nb-37.5 at. %Si, as in Figure 3. The presence of eutectic in the niobium-enriched region in the Nb-Si diagram (at a content of 18.7 at. %Si) makes it possible to obtain a composite structure in such alloys using directional solidification [18–20].

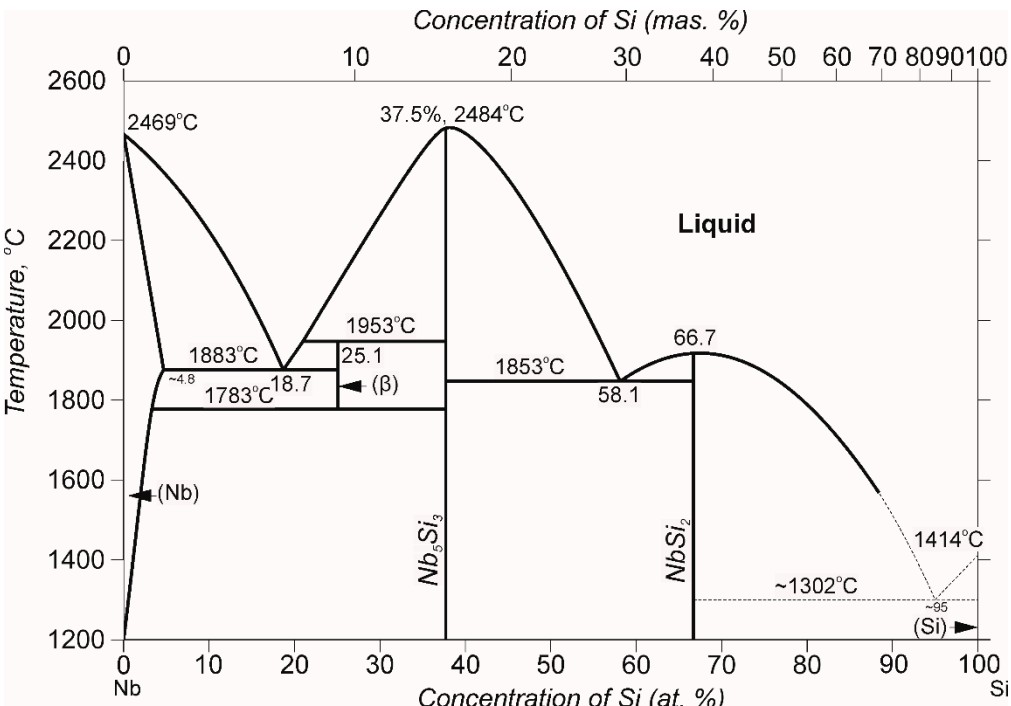

**Figure 3.** Schematic binary Nb-Si phase diagram. Taken from Ref. [12] with permission from ASM. Copyright 1986, ASM.

With the directed crystallization of Nb–Si-based alloys near eutectic composition, a composite structure arises consisting of columnar grains of a solid solution based on Nb, elongated along the growth axis, and continuous fibers or plates of the second phase: silicides, which are also oriented along the growth axis. This type of alloy includes MASC (metal and silicon composite), developed by the American company General Electric, which has the following chemical composition: Nb—25Ti—2Cr—8,2Hf—16Si—2Al (at. %), and its $T_m$ = 1760 °C. The eutectic of the MASC composite consists of a solid solution containing Nb, Si, Hf, Al, Cr and Ti, and silicides Me$_5$Si$_3$ and Me$_3$Si, which are the main strengthening phase of the composite structure.

Eutectic HRAs reinforced with intermetallic compound include (Cr-Ta)-based alloys, in which the eutectic structure consists of a solid solution based on chromium and the strengthening Laves phase MCr$_2$, where M = Nb, Ta and Zr [21,22]. A schematic phase diagram in binary alloys of this type is shown in Figure 4.

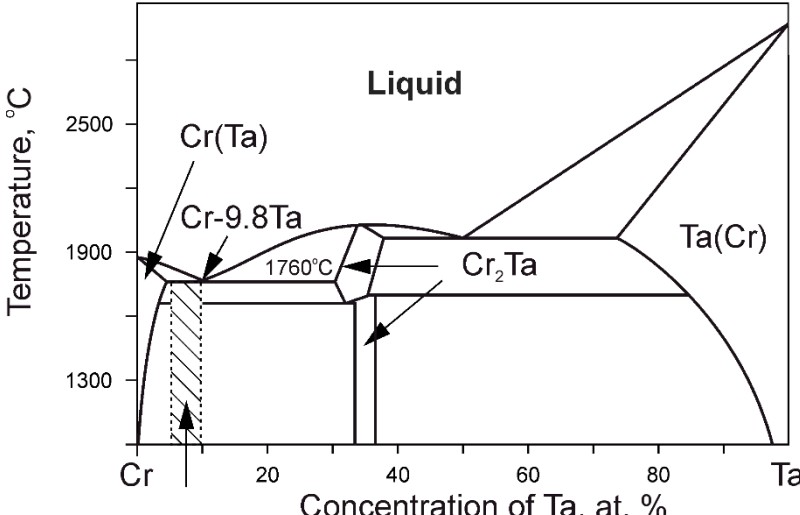

**Figure 4.** Cr-Ta binary alloy schematic phase diagram. Taken from Ref. [12] with permission from ASM. Copyright 1986, ASM.

The shaded area in the diagram of Figure 4 is close to the concentration range of the known alloy Cr–(6-10)Ta–(3-6)Mo–(0.2-1.0)Ti–(0.5-3.0)Si–(0.05-0.20)La\Ce (at.%), which has high mechanical properties. This alloy has a melting point of about 1750 °C, which is significantly higher than the $T_m$ of Ni-based HRAs. The creep resistance of the eutectic alloy Cr–8Ta–5Mo–0.5Ti–0.01Ce at a temperature of 1000 °C under a load of 138 MPa is at the level of properties of Ni-based HRAs with a single-crystal structure. In addition, the alloys of the Cr–$Cr_2Ta$ system have good oxidation resistance at elevated temperatures in air. However, a significant disadvantage of these alloys is a tendency to brittle fracture at room temperature.

HRAs reinforced with carbide fibers are CoTaC and NiTaC alloys obtained by directional solidification [23–26]. These alloys are based on pseudo-binary-phase equilibrium diagram (Co,Ni)–(Ta,Nb)C with eutectic, what makes it possible to obtain a composite metal–carbide structure consists of columnar grains in matrix elongated along the growth axis, in which "whiskers-like" crystals of monocarbides are located. The high stability of the composite structure in CoTaC alloys is ensured by the formation of special grain boundaries in the matrix and special Ni\TaC interface boundaries, which are characterized by low values of surface energy and diffusion permeability.

The "whiskers-like" fibers of the strengthening phase in such alloys have near theoretical strength, which ensures the production of a composite structure with uniquely high strength values and fatigue resistance at high temperatures. The main disadvantage of these alloys, which hinders their industrial application, lies in the technological field: the formation of a composite structure in them occurs during crystallization at a very slow rate.

### 2.3. Stability of the Polyphase Structure of HRAs: Kinetics of the Diffusion Coarsening Process

The matrix surrounds on all sides the particles of second phase, a typical structure for traditional heat-resistant Ni-based alloys. Effective dispersion hardening of the Ni-based HRAs is provided by optimal parameters ($\gamma$-$\gamma'$) microstructure, among which the most important are the particle size of the strengthening $\gamma'$-phase in the $\gamma$-matrix of the HRA, their number and morphology. However, during operation, diffusion coarsening of a given microstructure and deterioration of heat-resistant characteristics occur. To describe the kinetics of the diffusion coarsening process, the Lifshitz–Slezov–Wagner (LSW) model can be used [27,28], which describes the growth of large particles due to the dissolution of small ones over long times, as in Figure 5 [29].

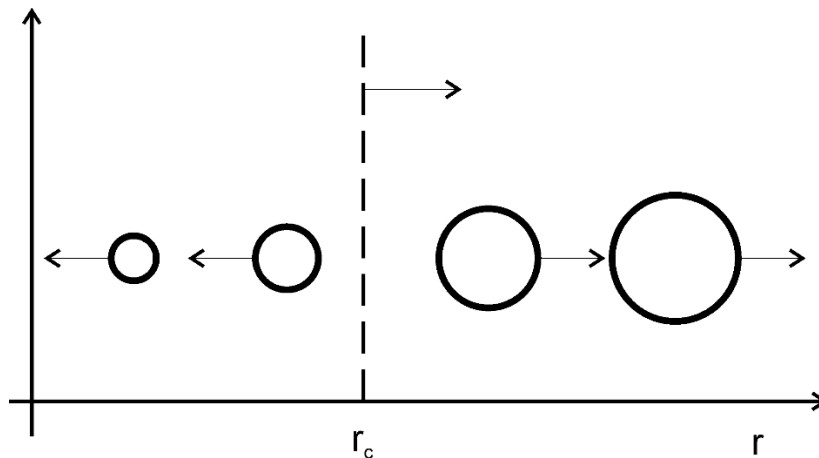

**Figure 5.** Diffusion coarsening scheme. Big particles grow, small particles dissolute. Open access from Razumovskii I.M. et al. [29].

According to the theory of diffusion coagulation, the average particle size r³ increases in proportion to the volume diffusion coefficient $D$ and time $t$:

$$r^3 = r_0^3 + \frac{4}{9}D\beta t;\tag{1}$$

where $\beta$ is the surface tension of the $\gamma/\gamma'$ interphases, $r_0$ is the initial particle size. Thus, it is important to emphasize that the stability of the $(\gamma\text{-}\gamma')$ microstructure will be controlled by the diffusion permeability of the $\gamma$ matrix.

In single crystals of Ni-based HRAs with a growth axis of ‹100› at high temperatures under the action of a tensile load a morphological transformation of a microstructure with isolated particles of the $\gamma'$ phase in the $\gamma$ matrix into a lamellar $\gamma\text{-}\gamma'$ structure, which is called a raft structure, occurs [15,16]. The presence of a raft structure in alloys leads to the need to evaluate the kinetics of the process of its diffusion coarsening at high temperatures.

A model of diffusion coarsening of lamellar structures is proposed in [30]. The model assumes that there are specific defects in the plates of the lamellar structure, namely, the holes which are filled with the material of the adjacent phase, as in Figure 6. According to this model, the coarsening of the lamellar structure at elevated temperatures occurs because of the diffusive growth of the hole-like defects. This process causes degradation of the initial structure, which loses its heat resistance. The change in the pore radius over time depends on the diffusion and structural parameters of the system and is determined by the equation

$$\frac{\mathrm{d}r}{\mathrm{d}t} = A\left(\frac{2}{h} - \frac{1}{r}\right),\tag{2}$$

where

$$A \cong \frac{c_\gamma}{c_{\gamma'} - c_\gamma}\frac{\beta\Omega}{h^2}\frac{(D_v l + D'\delta)}{RT}.\tag{3}$$

Here $l$ is the $\gamma$-layers thickness, $c_{\gamma'}\text{-}c_\gamma$ is the difference between diffusing component concentrations in $\gamma'$ and $\gamma$ phases, $D_v$ is the bulk diffusion coefficient in the $\gamma$ phase, $\beta$ is the surface tension of the $\gamma/\gamma'$ interphases, $\delta$ is $\gamma/\gamma'$ interphase boundaries width, $D'$ is the diffusion coefficient along these boundaries, $\Omega$ being the atomic volume. The model allows one to estimate the time during which the defect radius reaches a critical size, after which the rupture of the object may occur.

Model of diffusion coarsening of fibers in the composite structure of eutectics was proposed by Cline in [31]. It is assumed that during directed crystallization, fluctuating changes in the shape and diameter of fibers occur, which can activate two coarsening mechanisms: two-dimensional coagulation in L-S-W and spheroidization of fibers. To evaluate the kinetics of the coarsening process of a composite structure, it is necessary to

know the values of the diffusion coefficients of the components in the volume and at the phase boundaries [32].

(a)

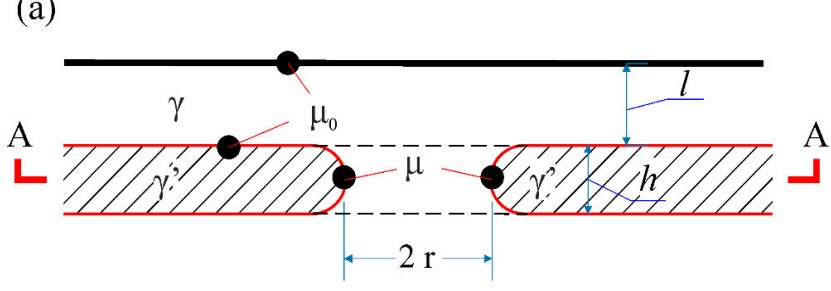

(b)

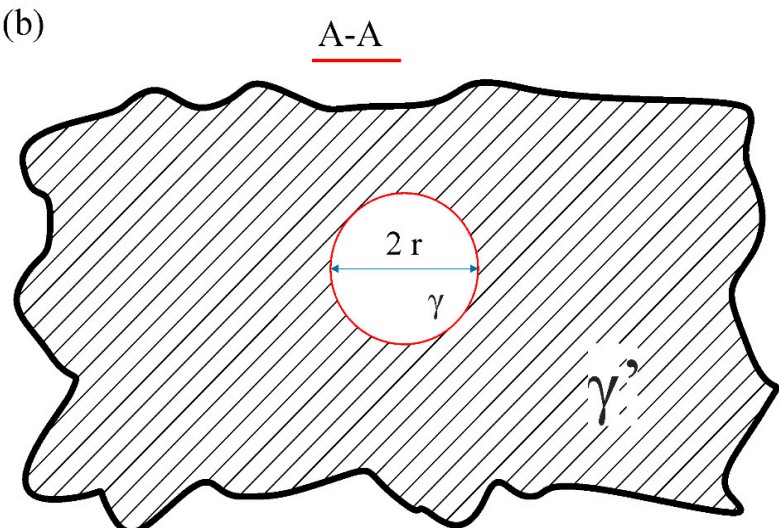

**Figure 6.** Schematic representation of a hole-like defect in the plate of the γ′-phase; (**a**,**b**) the sections perpendicular and parallel to the layers, accordingly. Open access from Razumovskii I.M. et al. [30].

### 3. Approaches to the Development of HRAs Based on Refractory Metals

Let us consider two directions of development of refractory HRAs, which seem to be the most promising.

#### 3.1. Alloys Based on Noble Metals

Among refractory HRAs, first, alloys based on noble metals should be noted, for which there is still an insufficiently fully studied resource for increasing heat resistance. Indeed, HRAs based on Pt and Ir used in industry are solid solutions, and they are strengthened by a solid-solution hardening mechanism [8]. However, a more effective approach to HRAs strengthening is the formation of a structure such as the (γ-γ′) microstructure of nickel HRAs [1–4], which uses a precipitation hardening mechanism. Therefore, it can be expected that the implementation of the mechanism of precipitate strengthening in Pt and Ir-based alloys will increase the heat resistance of these alloys.

Among the equilibrium diagrams of binary alloys based on Pt and Ir, we can distinguish several promising systems in which (γ-γ′) microstructures are formed: these are Pt-Al, Pt-Sc, Ir-Nb, Ir-Hf. However, among them, only in the Pt-Sc system, during alloying, there is no decrease in the melting temperature of Pt, which is favorable for performance. In all other systems, alloying lowers the melting point of Pt and Ir, as, for example, it occurs in alloys of the Pt-Al system [12,33,34], Figure 7.

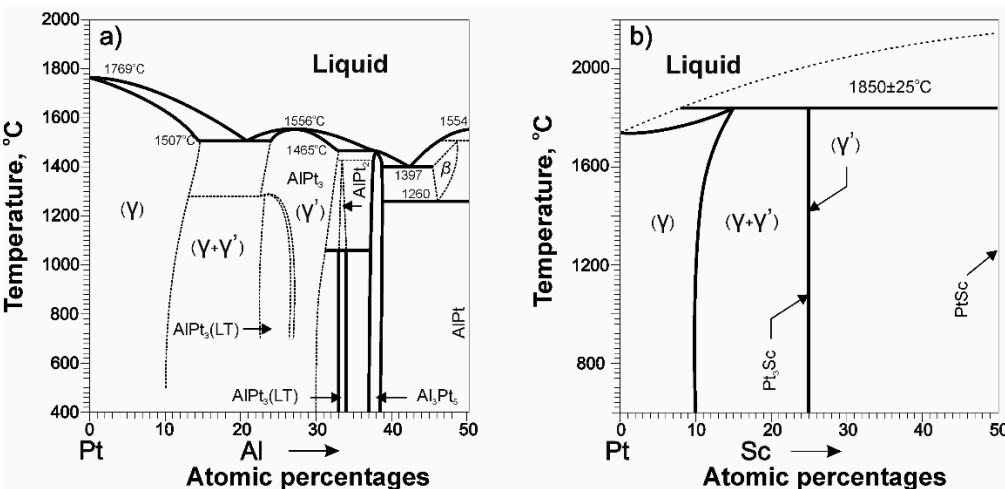

**Figure 7.** Phase diagrams schemes of binary alloys: (**a**) Pt-Al; (**b**) Pt-Sc. Taken from Refs. [12,34] with permission from ASM and Elsevier. Copyright 1986, ASM; 1980, Elsevier.

When choosing a system for alloying advanced HRA based on Pt and Ir with a (γ-γ′) microstructure, it should be borne in mind that the γ′-phase in such alloys must have high thermodynamic stability, determined by the enthalpy of formation of the compound $\Delta H_f$ [35,36]. This conclusion follows from the results of the theoretical evaluation of the $\Delta H_f$ values of many chemical compounds, which show that if we exclude noble metal compounds and silicides from consideration, the most stable is the intermetallic compound $Ni_3Al$ ($\Delta H_f = -0.49$ eV\atom)–the main strengthening phase of Ni-based HRAs. In alloys based on Pt and Ir, the most stable compounds are $Pt_3Sc$ ($\Delta H_f = -1.06$ eV\atom) and $Ir_3Hf$ ($\Delta H_f = -0.88$ eV\atom).

The idea to design a Pt-based HRAs with a (γ-γ′) microstructure similar to Ni-based superalloys is currently the subject of active research with the most widely studied alloys of the Pt-Al system [37–39]. Note, however, that Pt-Al alloys have several obvious disadvantages in terms of the high-temperature HRAs discussed here. (A) The introduction of Al significantly reduces the melting point of Pt, as in Figure 7a; that is, for this system in the region of existence of (γ-γ′)-phases, the criterion Tm ≈ 2000 °C is not met. (B) The $Pt_3Al$ compound is unstable: when cooled, a phase transformation occurs in it, which can have a negative effect on mechanical properties [40]. Nevertheless, some results of the study of Pt-Al alloys are important and are of interest for analyzing the state of the entire family of (Pt,Ir)–M alloys.

The Pt-Sc system is of particular interest for the development of advanced HRAs based on noble metals. A theoretical study of Pt-Sc alloys as the basis of promising HRAs was carried out in [41]. Alloying Pt with Sc, unlike Al as in Figure 7, increases the melting point of Pt, which is favorable for heat resistance. The $Pt_3Sc$ compound is characterized by high thermodynamic stability, determined by the enthalpy of formation of the compound $\Delta H_f = -1.06$ eV\atom) (for the $Ni_3Al$ intermetallic compound $\Delta H_f = -0.49$ eV\atom). The parameters of the Pt and the $Pt_3Sc$ compound crystal lattices are similar: the misfit value is less than 1%. This indicates the possibility of the formation of coherent phase boundaries γ\γ′ with low energy, as is observed in traditional Ni-based alloys.

Figure 8a,b show the results of first-principal calculations of elastic constants and elastic modulus both in disordered Pt-Sc alloys and for the ordered $Pt_3Sc$ compound. To theoretically assess the tendency of Pt-Sc alloys to brittle fracture according to the Pugh criterion, the ratio of the shear modulus to the bulk modulus G/B was calculated, which is usually less than 0.5 for plastic materials. It was found that the G/B values vary from 0.21 for pure Pt to 0.24 for the Pt-12.5 at. % Sc alloy, which indicates a high probability of ductile behavior of Pt–Sc alloys. The G/B ratio for the $Pt_3Sc$ compound is 0.3, which predicts its high ductility, unlike many other intermetallic compounds.

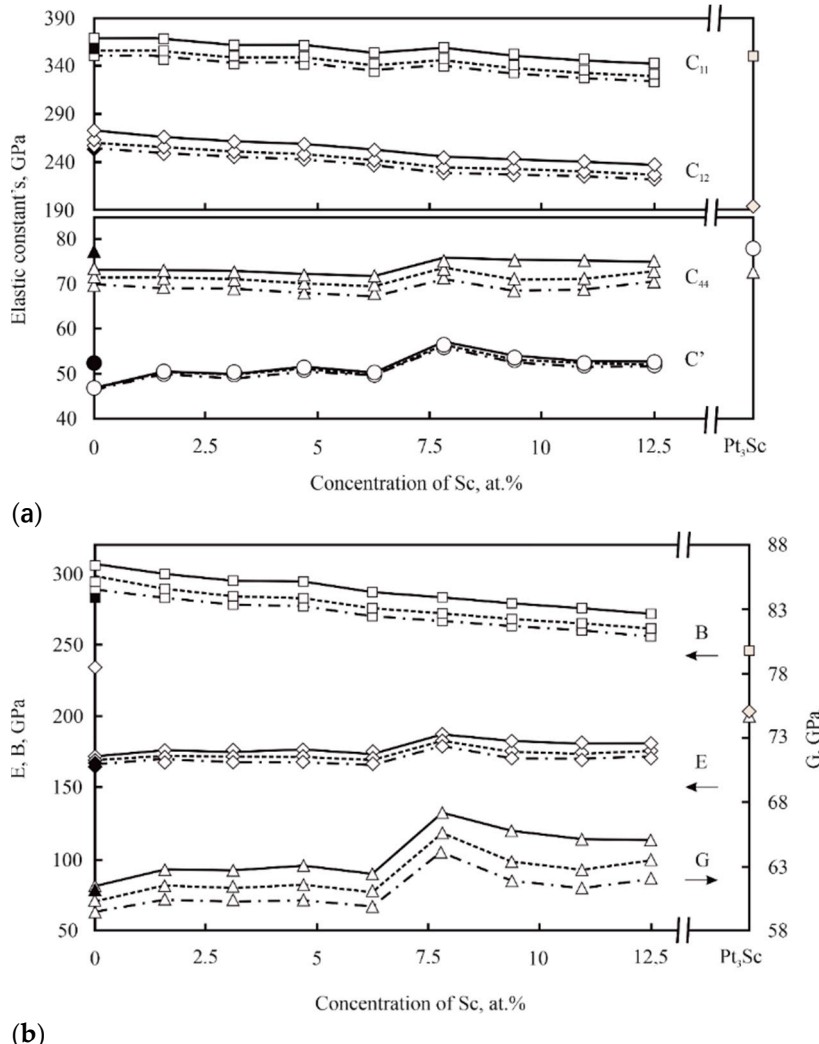

**Figure 8.** Calculated values of elastic constants (**a**); the bulk modulus B (open squares), Young's modulus E (open diamonds), and shear modulus G (open triangles) (**b**) in Pt-Sc alloys and Pt$_3$Sc compounds; solid and dotted lines represent the results of calculations using different approximations. Taken from Ref. [41] with permission from Elsevier. Copyright 2008, Elsevier.

Thus, based on the results of a theoretical study, the authors of [41] conclude that Pt-Sc alloys with ($\gamma$-$\gamma'$) microstructure that combine good strength and ductility and have a high melting point (the Pt$_3$Sc compound is stable up to a temperature of 1850 °C) are promising for creating advanced high-temperature HRAs.

Of even greater practical interest are the alloys of the Pt-Sc system alloyed with Ir. Indeed, the binary Pt-Ir equilibrium diagram is characterized by unlimited solubility. It is also known that Pt doping with Ir causes a significant increase in short-term and long-term strength due to solid-solution hardening while maintaining high values of elongation [8,9].

(Pt,Ir)-based HRAs with a ($\gamma$-$\gamma'$) microstructure will be of great interest for practical use if the stability of the ($\gamma$-$\gamma'$) microstructure with respect to the process of diffusion coarsening of the $\gamma'$-phase particles in a Pt and Ir-based matrix is high, comparable to that in Ni-based HRAs.

The relationship between the values of the diffusion permeability of the $\gamma$-matrix in the HRAs based on Ni, Pt, and Ir can be approximated by comparing the values of the self-diffusion coefficients in these elements. Using the values of self-diffusion coefficients to study the influence of various factors on the structural stability of alloys is an effective approach that is used in Ni-based HRAs to assess the durability and service life [14].

To directly compare the self-diffusion coefficients in Ni, Pt, and Ir, it is necessary to have their values at the same temperatures. However, experimental data for Ir [42] were obtained at temperatures, located above the melting points of Ni and Pt, so a direct comparison of the diffusion parameters of all three metals is impossible. Therefore, we will limit ourselves to comparing the values of the self-diffusion coefficients in Ni [32] and Pt [43] at the same temperatures given in Table 1.

**Table 1.** Values of self-diffusion coefficients D for Ni and Pt in the coinciding temperature range 1523–1670 K.

| T, K | 1523 | 1573 | 1623 | 1670 |
|------|------|------|------|------|
| Ni, D, $m^2/s$ | $3.0 \times 10^{-14}$ | $6.2 \times 10^{-14}$ | $1.2 \times 10^{-13}$ | $2.2 \times 10^{-13}$ |
| Pt, D, $m^2/s$ | $5.9 \times 10^{-15}$ | $1.2 \times 10^{-14}$ | $2.3 \times 10^{-14}$ | $4.2 \times 10^{-14}$ |

Table 1 shows that the diffusion permeability of Pt at all these temperatures is about 5 times lower than Ni; that is, the structural stability potential of the matrix of Pt alloys is greater than that of Ni-based alloys.

### 3.2. Alloys Based on Many Refractory Metals

Currently, the possibility of development HRAs with high melting temperatures based on many refractory metals (multi-principal element alloys) is being actively investigated [44–47].

Such alloys are often called "high-entropy alloys (HEAs)" because they are characterized by relatively large values of configuration entropy. For a long time, it has been considered that one of the key characteristics of HEAs is sluggish diffusion, which could be a very favorable factor in terms of heat resistance. However, the results of numerous studies of diffusion phenomena in HEAs—vacancy states [48] and self- and inter-diffusion—have shown that in most HEAs the phenomenon of sluggish diffusion is not observed [49,50]. At the same time, it should be emphasized that in most works HEAs with a FCC lattice have been studied; diffusion phenomena in refractory HEAs with a BCC lattice, which are of the greatest interest to us, have not yet been studied enough.

The matrix of such alloys is a solid solution based on many refractory metals which can be strengthened either by an intermetallic or silicide phases. However, most refractory alloys based on transition metals have a BCC lattice, which is characterized by a tendency to brittle fracture.

Nevertheless, in some cases, the tendency to brittle fracture of alloys with a BCC lattice can be minimized. The rule of ductility of refractory metals was formulated by Savitsky E. M. [51]. According to this rule, the greatest ability to plastic deformation, that is, to change the shape and size of a solid without destruction, is possessed by high-temperature modifications of all polymorphic metals in the BCC or FCC structures of which the atoms are characterized by a completely metallic bond. Such plastic refractory metals with polymorphic transformation include Ti, Zr, and Hf, in which the high-temperature BCC phase transforms into a low-temperature modification with a hexagonal close packed (HCP) lattice. It can be assumed that the ductility rule will be fulfilled not only in pure metals with polymorphic transformation, but also in alloys of these metals. Binary diagrams involving Ti, Zr, and Hf show that all systems are characterized by unlimited solubility of elements in the solid state and the presence of a high-temperature phase with BCC lattice, which, when cooled, transforms into HCP modification. All this indicates the possibility of using three-component solid solutions of the system (Ti, Zr, Hf) with polymorphic transformation as a matrix of advanced high-temperature HRAs.

The disadvantages of solid solutions of the system (Ti, Zr, Hf) as the matrix phase of HRAs is polymorphic transformation, which will occur when cooled from the expected operating ($\sim$0.75 $T_m$) temperatures. This disadvantage can be minimized by additional alloying of the alloys of the system (Ti, Zr, Hf) with isomorphic stabilizers of the BCC phase, which in Ti include V, Mo, Nb, Ta. Phase equilibrium diagrams in alloys (Ti, Zr, Hf)–(V, Mo, Nb, Ta) show that in alloys with V and Mo there is no complete solubility of the

components in the solid state. Therefore, V and Mo should be excluded from the number of main candidates for improving the characteristics of (Ti, Zr, Hf)-based alloys. Thus, solid solutions of the system (Ti, Zr, Hf) + (Ta, Nb) with a BCC lattice can be considered as a matrix of advanced refractory HRAs.

The results of experimental studies of the structure and mechanical properties of the equiatomic alloy TiZrHfNbTa obtained by traditional metallurgy show the formation of a single-phase BCC structure in which a small amount of low-temperature HCP modification can be found [52–55]. This alloy is well deformed at room temperature during tests of mechanical properties for compression and tension, which indicates the possibility of cold rolling of workpieces. The structure and properties of sheets obtained by rolling workpieces with a compression ratio of 86.4% were studied both in the initial state, immediately after rolling, and after annealing. It was found that the BCC phase with a typical microstructure of the deformed state was preserved in the deformed samples. After annealing at 1000 °C and 1200 °C, the BCC phase was mainly preserved in the samples; however, a small number of particles of the second phase were formed. During annealing, complete recrystallization of the deformed structure of the BCC phase and the formation of large equiaxed grains occurred; after annealing at 1000 °C, the following mechanical properties were obtained: $\delta = 9.7\%$, $\sigma_{0.2} = 1262$ MPa.

To produce alloys of the TiZrHfNbTa system, it is possible to use not only methods of traditional metallurgy (electric arc melting, etc.), but also powder metallurgy [56,57]. Thus, the available data show that alloying of three-component alloys (Ti, Zr, Hf) with strong BCC phase stabilizers (Nb, Ta) leads to the preservation of such an amount of stable BCC phase in the alloy at room temperature that can provide acceptable ductility of the alloys. The strength of the TiZrHfNbTa alloys with an "almost homogeneous" structure at room temperature is about $\sigma_B = 1000$ MPa, that is, close to the strength of some Ni-based HRAs.

Considering the problem of choosing the chemical composition of the matrix of alloys based on many refractory metals solved, approaches to matrix strengthening by the second phase should be studied, which is necessary to ensure long-term strength at high temperatures. The strengthening of considered alloys can be provided by eutectic structures consisting of a metal solid solution and intermetallic or silicide phases, the formation of which occurs with additional alloying.

For example, it is known that the formation of intermetallic Laves phases in refractory alloys based on many components occurs when alloyed with chromium. The authors [58] found the Laves phase $Cr_2M$ in the equiatomic alloy TiZrHfNbCr, in which Ta was completely replaced by Cr in the considered composition (Ti-Zr-Hf-Ta-Nb). An attractive feature of such alloying is a decrease in the density of the alloy due to the replacement of "heavy" Ta ($\rho = 16.6$ g/cm$^3$) with "light" Cr ($\rho = 7.14$ g/cm$^3$); however, this effect is accompanied by a negative factor—a decrease in melting temperature. The combination of the solid solution BCC with the Laves phase provides a relatively high level of compressive strength $\sigma_B = 1900$–2150 MPa, although the ductility is only a few percent. Nevertheless, the authors [58] predict good ductility of the BCC solid solution in the TiZrHfNbCr alloy due to the favorable ratio of elastic modules (Pugh criterion): B/G = 3.0 (bulk modulus B\shear modulus G).

Solid solution strengthening with silicides is another promising direction for the development of HRAs based on many refractory metals [59]. In work [59], a cast alloy based on several refractory metals with silicon of the composition HfNbTiVSi0.5 was investigated. The eutectic structure of the castings consisted of a BCC solid solution enriched with Hf, Nb, Ti, V and depleted Si, and a multicomponent silicide (Hf, Nb, Ti)-Si. The alloy had a density of $\rho = 8.6$ g/cm$^3$, the values of the yield strength of the alloy during the compression test $\sigma_{0.2}$ were 1399 MPa, 875 MPa, and 240 MPa at temperatures of 20 °C, 800 °C, and 1000 °C, respectively.

However, the most interesting approach to the development of structural alloys based on many refractory metals is to obtain a structure in which the BCC matrix (multicomponent solid solution) is reinforced with particles of the ordered phase B2 [60,61]. This leads to the

formation of a coherent BCC–B2 microstructure such as the γ-γ′ microstructure in Ni-based HRAs, which can have high structural stability and provide high long-term strength.

The formation of the B2 phase in BCC solid solutions of alloys based on many refractory metals occurs when alloyed with Al [61], and this behavior is not typical for binary M–Al alloys. It should be noted that when Al is added, there is a significant decrease in the melting temperature of binary alloys M–Al (except for Ti). The melting point value is especially important for the development of a new generation of HRAs with melting temperatures above 2000 °C and potential operating temperatures of 1500 °C. That is why it is important to search for the chemical composition of HRAs based on many refractory metals with a coherent BCC–B2 microstructure and a melting point above 2000 °C.

A significant influence on the creep resistance of HRAs at high temperatures is exerted by the characteristics of the cohesive strength of alloys—the cohesion energy of the solid solution and the cohesive strength of the grain boundaries (GB) [62]. The parameters of the cohesive strength of alloys depend on the chemical composition, which can be adjusted to increase heat resistance. This approach has been successfully used to develop promising HRAs based on Ni, Ti, and Cr.

At first glance, it seems that such an approach is not applicable to improve the properties of HRAs based on many refractory metals, each of which is characterized by a high value of the cohesion energy [63], which is difficult to increase by alloying. However, to increase the cohesive strength of GBs in refractory alloys, this approach can be used.

The cohesive strength of the interfaces in alloys, according to the Rice, Thomson, and Wang model [64,65], is characterized by the work of the separation of GB. The ideal work of separation $W_{sep}$ is the reversible energy needed to separate two grains, which is equal to:

$$W_{sep} = 2\gamma_{fs} - \gamma_{gb}, \tag{4}$$

where $\gamma_{fs}$ and $\gamma_{gb}$ are the energies of the respective free surface and GB per unit area.

Thus, the cohesive strength of GB is controlled not only by the GB, but also by the free surface. The value of the cohesive strength of GB effects both the creep resistance of the alloy at high temperatures and the tendency toward GB brittle fracture at room temperature. A theoretical analysis of the tendency of BCC metals to brittle fracture along the GB from the standpoint of the cohesive strength of the boundaries was carried out in [66,67]. It has been shown that among the BCC metals W and Mo are characterized by a high tendency toward GB fracture due to the specific ratio of the values of $\gamma_{fs}$ and $\gamma_{gb}$ which determines the cohesive strength of the GB. The segregation of the alloying element to interfaces can change this ratio increases the ductility in refractory metals. For example, GB brittleness was also observed in a refractory alloy based on many elements (HEA). Nb25Mo25Ta25W25 [68], therefore, when developing new HRAs, it is possible to increase the ductility of the BCC matrix by special alloying with "grain boundary" elements [3]. An increase in the cohesive strength of GBs because of special alloying can also contribute to an increase in creep resistance. Features of the effect of the entropy factor on GB segregation in alloys can be found in [69].

Theoretical methods, including first-principles calculations, are very effective for studying the segregation ability of elements [70]. This approach proved to be very useful for studying the GB segregation of elements in traditional Ni, Cr, and Ti alloys [71]. Recently, theoretical methods have been developed to study the segregation of elements in alloys based on many components and used to calculate the surface segregation in FCC random Fe70Cr20Ni10 and FeMnCoCrNi alloys [72]. Similar methods can be developed and applied to the study of segregation in alloys with a BCC lattice, and not only on the free surface, but also at the GBs. This will make it possible to find elements that strengthen GB in BCC alloys based on many components and use them as low-alloying additions by analogy with traditional HRAs.

## 4. Final Notes

In this paper, approaches to the development of two groups of advanced refractory HRAs are considered: alloys based on Pt-Ir-Sc (I) and alloys based on many refractory non-noble metals (II).

Alloys (I), despite their high cost, can be used, for example, for the manufacture of critical space craft thruster components [73]. It is assumed that for Pt-based alloys, the best way to increase the heat resistance characteristics is a combination of solid-solution hardening and precipitation-hardening mechanisms. Effective solid-solution hardening of Pt-based alloys can be achieved by doping Ir, and this approach has been well-studied experimentally. To implement the precipitation-hardening mechanism, solid solutions based on (Pt-Ir) can be alloyed with Sc, the introduction of which leads to the formation of the $Pt_3Sc$ compound. As a result, a two-phase $(Pt,Ir)$-$Pt_3Sc$ microstructure similar to the unique $\gamma$-$\gamma'$ microstructure of Ni-based HRAs can be formed.

Alloys based on many refractory metals are of particular interest. Multicomponent alloying system of such alloys with a high concentration of each of the alloying elements provides an unusually large variability for structural alloys in changing the chemical composition and physicochemical properties. Multicomponent HRAs have to meet the following criteria: a high melting point of $\geq 2000\ ^\circ$C, acceptable ductility at room temperature, and the ability to form a heat-resistant heterophase structure. It is shown that solid solutions of the system (Ti, Zr, Hf, Ta, Nb) with a BCC lattice can be considered as a matrix of new refractory HRAs.

To increase the creep resistance of the matrix at high temperatures, it is advisable to strengthen the GBs with elements that segregate to the GBs and increase the cohesive strength of the boundaries. For alloys of a given chemical composition, such elements can be established using theoretical methods including, for example, first-principles calculations.

To create a heat-resistant heterophase structure in alloys based on many refractory metals, it seems that the most interesting approach is the strengthening of the BCC matrix by particles of an ordered BCC phase (B2). To do this, it is necessary to find chemical elements that, when doped, will cause the diffusion decomposition of a solid solution with a BCC lattice, the formation of an ordered B2 phase while maintaining a high melting point $\geq 2000\ ^\circ$C. Apparently, this problem in relation to alloys based on many refractory metals has not yet been solved.

For the long-term operation of advanced alloys based on many refractory metals at high temperatures, it will be necessary to protect the surface of the parts from oxidation and other factors of the aggressive external environment. For such a function, alloys based on (Pt-Ir), which are used to protect the surface of heat-resistant Ni-based alloys, can be suitable, and when developing new compositions of such alloys considered in this paper, their protective abilities can be taken into account. Thus, it is advisable to develop alloys of families I and II, as discussed above, within the framework of a unified approach to create a new generation of structural and functional materials.

**Author Contributions:** I.R. and B.B. formulated the idea of this work and wrote all its sections except Section 2.3, which were written by M.R. A special contribution to the preparation of the manuscript for publication was made by M.R. All authors have read and agreed to the published version of the manuscript.

**Funding:** This work was supported by the Russian Foundation for Basic Research, Grant No. 20-03-00387.

**Conflicts of Interest:** Igor Razumovskii is chief researcher at the Joint Stock Company "Kompozit". Boris Bokstein is a full professor at the National University of Science and Technology (MISIS). Mikhail Razumovsky is a postgraduate student at the National University of Science and Technology (MISIS).

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
