# Peer review of "Approaches to the Development of Advanced Alloys Based on Refractory Metals"

_encyclopedia, doi:10.3390/encyclopedia3010019_

Round 1

Reviewer 1 Report (New Reviewer)

Interesting paper on the development of alloys resistant to high
temperatures. The part dedicated to high entropy materials deserves
special consideration.

Author Response

Dear Editors,

We have considered the comments of our referees and suggest the following.

Reviewer's comments:

Reviewer 1

  1. Interesting paper on the development of alloys resistant to high
    temperatures. The part dedicated to high entropy materials deserves
    special consideration.

Answer.

We prefer to leave the materials of section 3.2. (Alloys based on many refractory metals) in the form we have proposed, since more detailed information about HEAs can be found in the references we have indicated [42,43,44,45] as well as some of our other works, for example, 1) Razumovskii I.M., et al. Alloys based on many refractory metals—next generation of high‐temperature composite materials // Konstrukts. Composite. Mater. 2019, № 1; 2) Physics and chemical factors of heat resistance of alloys based on many refractory metals: an introduction to the problem // Konstrukts. Composite. Mater. 2022, № 3, DOI 10.52190/2073-2562_2022_3_25.

Reviewer 2 Report (New Reviewer)

The manuscript analyzes the directions of development of refractory metal-based heat-resistant alloys. The organizational characteristics of HRAs suitable for formation in promising alloys are considered. The factors influencing the diffusion coarsening of the hardened phase on the tissue stability are discussed. Two groups of alloys are considered to be the most promising refractory metal-based HRAs. This study is very good, but there are still some minor problems, that the author should explain and improve.

1. The development of new alloys based on refractory metals can be carried out in a variety of ways, including alloying, heat treatment, surface modification, and the use of computational models and simulations. This manuscript only included alloying and computer modeling.

2. The conclusion of the summary section of the abstract on the second group of alloys should be placed at the end.

3. In this manuscript, the addition of Sc elements to Pt-Ir binary alloy materials is recommended. In addition to this, many scholars study the addition of Hf , Zr, Y and other elements. So, what is the difference in comparing the effects produced by these elements on the alloy? Which alloying element is more suitable?

4. The content of subsection on 2.3 describes the model for diffusion coarsening. In addition to the LSW model and Cline model introduced in this manuscript. There are also the Cahn-Hilliard (CH) model, Allen-Cahn (AC) model, Mullins-Sekerka (MS) model, etc. What are the differences between these models?

5. Please standardize the format of the documentation. For example, Ref. 58.

Author Response

Dear Editors,

We have considered the comments of our referees and suggest the following.

Reviewer's comments:

Reviewer 2

  1. The development of new alloys based on refractory metals can be carried out in a variety of ways, including alloying, heat treatment, surface modification, and the use of computational models and simulations. This manuscript only included alloying and computer modeling.

Answer.

We understand that the development of new alloys requires solving many problems in the field of physical metallurgy and technology (see, for example, our Russian Patents No. 2,348,724; № 2,344,190; № 2,344,195; № 2,348,724; № 2,353,691; № 2,361,011; № 2,361,012; № 2,620,405 and others). But we also know that scientifically based choice of the chemical composition and structure of the alloy is the basis of successful work. It is these issues that we are considering in our article.

  1. The conclusion of the summary section of the abstract on the second group of alloys should be placed at the end.

Answer.

The proposed form of presentation of the material in the abstract seems to us quite acceptable.

  1. In this manuscript, the addition of Sc elements to Pt-Ir binary alloy materials is recommended. In addition to this, many scholars study the addition of Hf , Zr, Y and other elements. So, what is the difference in comparing the effects produced by these elements on the alloy? Which alloying element is more suitable?

Answer.

We tried to answer this question by changing the text:

  1. 8, lines 249-251, it is written:

Among the equilibrium diagrams of binary alloys based on Pt and Ir, we can distinguish several promising systems in which (γ - γ’) microstructures are formed: these are Pt-Al and Pt-Sc [10,31,32], Figure 7.

We offer an extended version of this fragment in the following form:

Among the equilibrium diagrams of binary alloys based on Pt and Ir, we can distinguish several promising systems in which (γ - γ’) microstructures are formed: these are Pt-Al, Pt-Sc, Ir-Nb, Ir-Hf. However, among them, only in the Pt-Sc system, during alloying, there is no decrease in the melting temperature of Pt, which is favorable for performance. In all other systems, alloying lowers the melting point of Pt and Ir, as, for example, it occurs in alloys of the Pt-Al system [10,31,32], Figure 7.

  1. The content of subsection on 2.3 describes the model for diffusion coarsening. In addition to the LSW model and Cline model introduced in this manuscript. There are also the Cahn-Hilliard (CH) model, Allen-Cahn (AC) model, Mullins-Sekerka (MS) model, etc. What are the differences between these models?

Answer.

We are considering the following task. It is assumed that in this HRAs the optimal microstructure of the types of standards for HRAs has been formed: isolated particles of the strengthening phase in the matrix or a composite structure.

When heated, which is unavoidable for the operation of HRAs, a process of diffusion coarsening of a given microstructure occurs, the kinetics of which controls the durability of the alloy. To evaluate the kinetics of this process, we propose to use the models described in this paper. Our experience shows that such an approach justifies itself for predicting the structural stability of HRAs (see, for example, Bokshtein, S. Z., Ginsburg, S. S., Razumovskii, I. M., Kishkin, S. T., B. Stroganov G. Autoradiography of interfaces and structural stability of alloys // Moscow, Izd. Metallurgy, 1987. - 270 p. (In Russian); Bokstein, B.S. Diffusion in Metals, 2nd ed. // Lenand: Moscow, Russian, 2019. – 248 p. (In Russian)).

  1. Please standardize the format of the documentation. For example, Ref. 58.

Answer.

The editorial office will help arrange the format of references before publication.

Reviewer 3 Report (New Reviewer)

The review article systematically summarized the compositions and microstructures of heat-resistant alloys. Based on the design concepts of traditional single principal element alloys and recently developed multi-principal element alloys, the existing alloy systems have been analyzed and discussed scientifically from the aspects of phase structure design and the relationship between components and microstructure. This review article provides positive guidance for composition design and microstructure regulation of heat-resistant alloys and refractory alloys. I suggest that this manuscript can be published in Encyclopedia.

Author Response

Dear Editors,

We have considered the comments of our referees and suggest the following. 

Reviewer's comments:

Reviewer 3

The review article systematically summarized the compositions and microstructures of heat-resistant alloys. Based on the design concepts of traditional single principal element alloys and recently developed multi-principal element alloys, the existing alloy systems have been analyzed and discussed scientifically from the aspects of phase structure design and the relationship between components and microstructure. This review article provides positive guidance for composition design and microstructure regulation of heat-resistant alloys and refractory alloys. I suggest that this manuscript can be published in Encyclopedia.

Answer.

Thank you so much for such an assessment of our work!

Round 2

Reviewer 2 Report (New Reviewer)

The manuscript analyzes the directions of development of refractory metal-based heat-resistant alloys. The organizational characteristics of HRAs suitable for formation in promising alloys are considered. The factors influencing the diffusion coarsening of the hardened phase on the tissue stability are discussed. Two groups of alloys are considered to be the most promising refractory metal-based HRAs. The author has made a reasonable explanation and the manuscript is available for acceptance.

  •  

Author Response

The manuscript analyzes the directions of development of refractory metal-based heat-resistant alloys. The organizational characteristics of HRAs suitable for formation in promising alloys are considered. The factors influencing the diffusion coarsening of the hardened phase on the tissue stability are discussed. Two groups of alloys are considered to be the most promising refractory metal-based HRAs. The author has made a reasonable explanation and the manuscript is available for acceptance.

Answer.

Thank you so much for such an assessment of our work!

This manuscript is a resubmission of an earlier submission. The following is a list of the peer review reports and author responses from that submission.

Round 1

Reviewer 1 Report

In this work, the authors review the refractory alloys with different components and strengthening mechanism used in the turbine machine. The proposal of adopting noble Pt-Ir-Sc alloys to manufacture the miniature heat-resistant parts of the thermocatalytic engines is of certain interest. Overall, there are many problems in this article, and the article should be carefully revised before possible publication.

Major comments:

- What is the purpose of this article? The outline and purpose of this article can be given at the end of introduction. 

- For variable “Tm” in line 32, “T” should be written in Italic style, and “m” should be written in subscript style. Similar issues exist for some other variables.

- The directional coarsening or so-called rafting of γ’ phase is a significant microstructure evolution for the precipitation-strengthening nickel-based alloy under uniaxial loading. It is suggested to discuss related studies here.

- What is the meaning of line 202?

- Lines 203 to 210 are repetitive in the article.

- The definition of the criterion Pugh in line 293 should be given in the article.

- What is the meaning of 60.2 = 1262 MPa in line 389?

- What is the meaning of ψ = 11% in line 423?

- Does the “strengthener” in line 131 mean “strengthening phase”?

- There are 17 self-cited references in this review paper. Though these references are related to the topic of this review, most of self-citations are not specifically discussed in the review. Any explanations?

Minor comments:

- Several short paragraphs could be combined together as a long paragraph. For example, three short paragraphs from lines 88 to 98.

- Variable r0 in Eq. 1 is not defined.

- The index of subfigures in figure 7 should be located in the same position.

- There should be no indentation in lines 180, 197, 198, and 453.

Author Response

Comment 1:

1) What is the purpose of this article? The outline and purpose of this article can be given at the end of introduction.

Author's response to the comment:

We fully agree with this remark and formulated the purpose of our article at the end of the introduction in the following form:

In this paper, the most promising approaches to the development of the newest HRAs based on refractory metals are analyzed. The features of the microstructure of HRAs, which provide high long-term strength of alloys, and the mechanisms of diffusion coarsening of different types of structures are considered. The design principles of refractory HRAs based on precious metals and high-entropy alloys   are proposed.

In addition, in the introduction, the following text is added after line 45:

Currently, the possibility of creating metallic HRAs with high melting temperatures based on many refractory metals (so-called "high-entropy alloys") is being actively investigated. The matrix of such alloys is a solid solution with a BCC lattice, which is characterized by a tendency to brittle fracture. Therefore, it is necessary to develop approaches to the choice of alloying system for such alloys with an acceptable deformation ability.

Comment 2:

 2) For variable “Tm” in line 32, “T” should be written in Italic style, and “m” should be written in subscript style. Similar issues exist for some other variables.

Author's response to the comment:

We made such an edit in the text of the entire article.

Comment 3:

3) The directional coarsening or so-called rafting of γ’ phase is a significant microstructure evolution for the precipitation-strengthening nickel-based alloy under uniaxial loading. It is suggested to discuss related studies here.

Author's response to the comment:

For a brief explanation of this issue, we have inserted the following text on page 7:

In single crystals of nickel HRAs with a growth axis of ‹100› at high temperatures, under the action of a tensile load, a morphological transformation of a microstructure with isolated particles of the γ’ phase in the γ matrix into a lamellar γ - γ’ structure, which is called a raft structure, occurs. The formation time of the raft structure is usually short compared to the durability of the alloy, therefore, the kinetics of diffusion coarsening of the raft structure largely determines the life of the alloy during operation.

Comments 4, 5:

4) What is the meaning of line 202?

5) Lines 203 to 210 are repetitive in the article.

Author's response to the comments:

We have deleted a few lines: they really repeat the above text.

Comment 6:

6) The definition of the criterion Pugh in line 293 should be given in the article.

Author's response to the comment:

It seems to us that the phrase given by us in the text of the article quite fully explains the meaning of the Pugh criterion:

«To theoretically assess the tendency of Pt-Sc alloys to brittle fracture according to the criterion Pugh, the ratio of the shear modulus to the bulk modulus G/B was calculated, which is usually less than 0.5 for plastic materials».

Comment 7:

7) What is the meaning of 60.2 = 1262 MPa in line 389?

Author's response to the comment:

Fixed: should be σ0.2 = 1262 MPa.

Comment 8:

8) What is the meaning of ψ = 11% in line 423?

Author's response to the comment:

We have removed this parameter from the text.

Comment 9:

9) Does the “strengthener” in line 131 mean “strengthening phase”?

Author's response to the comment:

Yes, you're right. We have replaced the word “strengthener” with “strengthening phase”.

Comment 10:

10) There are 17 self-cited references in this review paper. Though these references are related to the topic of this review, most of self-citations are not specifically discussed in the review. Any explanations?

Author's response to the comment:

For a long time, our team has been successfully engaged in the development of the scientific direction "Diffusion, interfaces and cohesive strength of high-temperature alloys". Some of the results obtained are given in our papers, which are cited in this article. Based on the results obtained, new structural alloys and technologies were developed (see, for example, Russian Patents No. 2,348,724; № 2,344,190; № 2,344,195; № 2,348,724; № 2,353,691; № 2,361,011; № 2,361,012; № 2,620,405 and others). It seems to us that the application of this approach will be useful in the development of new alloys based on refractory metals. Therefore, we would not like several our works to be excluded from the bibliographic list proposed by ourselves.

Other reviewer comments:

Comment 11:

- Several short paragraphs could be combined together as a long paragraph. For example, three short paragraphs from lines 88 to 98.

Author's response to the comment:

Agree. We have combined short paragraphs.

Comment 12:

- Variable r0 in Eq. 1 is not defined.

Author's response to the comment:

r0 is the initial particle size, we put it in the text.

Comment 13:

- The index of subfigures in figure 7 should be located in the same position.

Author's response to the comment:

We have excluded subfigures 7 (c,d) from the manuscript, and it seems to us that we have corrected this inaccuracy.

Comment 14:

- There should be no indentation in lines 180, 197, 198, and 453.

Author's response to the comment:

The changes have been made.

That Note: corrections made are highlighted in yellow in the manuscript.

Reviewer 2 Report

-        I would indicate Tm as Tm, and since you speak about a melting temperature in Celsius (“1450°C”) I suggest to explicitly state that that when you write “0.75 Tm” you are referring to melting temperature in K.

-        Please replace expressions like “Ni3Al” with “Ni3Al”

-        When discussing the stability of gamma-gamma prime system (section 2.3) a mention should be made about the phenomenon of rafting, which occurs at high temperature. The statement that “a given microstructure” can be maintained “for a long time under the influence of high temperatures and loads“ can’t hold, for example, for a Ni-based single crystal at temperatures close to 1000°C: even after a few hours, cubic gamma prime particles evolve into a different microstructure. Rafting is present also in Co-alloys (see https://www.science.org/doi/10.1126/science.1121738 and https://www.sciencedirect.com/science/article/pii/S1359646212000243#b0005, for example)

-        At row 159, what do you mean by “near theoretical strength”?

-        In row 202 some characters are corrupted and I couldn’t read them. Rows 203-206 are exactly the same as rows 198-201. The same applies to rows 207-210, which are the same as rows 211-214. I don’t really understand the meaning of “rapture of the object”.

-        I think that section 2.3 should face the issue of diffusion under applied stress. In the introduction section the focus is set on structural materials, which are required to sustain loads. It is well known that an applied stress interfere with diffusion (Earlier I mentioned rafting: without an applied stress, even at high temperatures it doesn’t occur for long times).

-        If possible, the summary at row 259 (and following) should include specific references for each claim

-        Fig. 9: Maybe you should indicate the activation energy used for calculating the “virtual” diffusion coefficients. At any rate, how significant is the calculation of diffusion coefficient above melting temperature? Wouldn’t it be easier to calculate the coefficient of Pt at lower temperatures? I mean, the performance of Ni alloys above their melting temperatures is obviously poor, even without comparing “virtual” and actual values.

-        Row 389 is unclear to me, I don’t understand “δ = 9.7%, Ϭ0.2 = 1262 MPa.” I guess they are elongation to fracture and 0.2% proof stress. Please use the subscript also in this case. (similar comment on row 396)

-        In section 3.2 you include Ti among refractory metals, but its melting temperature is about 1600°C, is it correct? I don’t understand why you mention (row 391) the use of powder metallurgy to produce this alloys, in my opinion this information looks somewhat uncorrelated to the rest of the paragraph. At row 396 there is again a symbol which is not defined, σB.

-        At row 412 you use “plasticity” but I guess that you are referring to “ductility”. I guess this happens a few times throughout the paper, so please verify that the term is appropriate.

-        Check the symbols in 421-423 (what is ψ?)

Author Response

Comments and Suggestions for Authors

Comment 1:

-        I would indicate Tm as Tm, and since you speak about a melting temperature in Celsius (“1450°C”) I suggest to explicitly state that that when you write “0.75 Tm” you are referring to melting temperature in K.

Author's response to the comment:

We agree. That's how we understand it. However, the discussion of this issue in the article seems superfluous to us.

Comment 2:

-  Please replace expressions like “Ni3Al” with “Ni3Al”

Author's response to the comment:

Done

Comment 3:

When discussing the stability of gamma-gamma prime system (section 2.3) a mention should be made about the phenomenon of rafting, which occurs at high temperature. The statement that “a given microstructure” can be maintained “for a long time under the influence of high temperatures and loads“ can’t hold, for example, for a Ni-based single crystal at temperatures close to 1000°C: even after a few hours, cubic gamma prime particles evolve into a different microstructure. Rafting is present also in Co-alloys (see https://www.science.org/doi/10.1126/science.1121738 and https://www.sciencedirect.com/science/article/pii/S1359646212000243#b0005, for example)

Author's response to the comment:

We have already given a brief explanation of this question in the answer to the 3rd comment of Reviewer 1. Indeed, the time of formation of the raft structure in different alloys may be different, but we rely on the results of our work presented in reference 31:

31.Mishin, Yu., Orekhov, N., Alyoshin, G., Noat, P., Razumovskii, I. Model of diffusion coarsening of the raft structure in single crystals of Ni-based superalloys // Materials Science and Engineering A. 1993, 171, 163–168. https://doi.org/10.1016/0921-5093(93)90402-Z.

Comment 4:

At row 159, what do you mean by “near theoretical strength”?

Author's response to the comment:

The theoretical strength of solids is the maximum possible stress that an ideal solid can withstand without rupture. The strength of real metals and alloys is significantly lower than the theoretical one. However, the strength of “whiskers” (single crystal fibers) is close to the theoretical value.

Comment 5:

-        In row 202 some characters are corrupted and I couldn’t read them. Rows 203-206 are exactly the same as rows 198-201. The same applies to rows 207-210, which are the same as rows 211-214. I don’t really understand the meaning of “rapture of the object”.

Author's response to the comment:

Thank you for that comment. Rupture is the time to rupture.

Comment 6:

-        I think that section 2.3 should face the issue of diffusion under applied stress. In the introduction section the focus is set on structural materials, which are required to sustain loads. It is well known that an applied stress interfere with diffusion (Earlier I mentioned rafting: without an applied stress, even at high temperatures it doesn’t occur for long times).

Author's response to the comment:

We do not agree with this remark: the issues of the influence of stresses on diffusion are not considered in this paper, as well as the diffusion aspects of rafting. For us, rafting is just one of the examples of lamellar structures in alloys. For example, in our work Kardashova S. et al. Diffusion Coarsening of the Lamellar Structure in Two-Phase Ti-47.5 at. % Al Intermetallic Alloy // Acta Metall. 1994, 42, 3341–3348. https://doi.org/10.1016/0956-7151(94)90466-9), the Ti-Al system is considered.

Comment 7:

-        If possible, the summary at row 259 (and following) should include specific references for each claim

Author's response to the comment:

If we understood correctly, Reviewer 2 has the following conclusions in mind:

  1. A heterophase and regular (γ - γ’) microstructure like in Ni-based HRAs is formed in Pt-Al alloys.
  2. Alloying of binary Pt-Al alloys with several transition metals-Ru, Cr, Ta, Ir, and Ti – can change the (γ - γ’) microstructure, which expands the ability to control the structure and properties of alloys by changing the chemical composition.

III.          At high temperatures above 1150°C, the strength of Pt-based alloys of the Pt-Al-Z alloying system (Z = Ti, Cr, Ru, Ta, and Re) exceeds the strength of the MAR-M247 Ni-based HRA. This indicates the potential use of Pt -based alloys with a (γ - γ’) microstructure at high temperatures that are not available for Ni- based HRAs due to the proximity of the test temperature to their melting point.

  1. The effect of strengthening of a γ-matrix by dispersed precipitation of the γ ' phase in Pt-based alloys is significantly greater than that of solid-solution hardening.

It seemed to us that these conclusions were clear and did not require explanation. However, we see that Reviewer 2 does not agree with this, so we simply exclude this information from our article. Along with this, we exclude Figure 7 (c,d).

Comment 8:

-        Fig. 9: Maybe you should indicate the activation energy used for calculating the “virtual” diffusion coefficients. At any rate, how significant is the calculation of diffusion coefficient above melting temperature? Wouldn’t it be easier to calculate the coefficient of Pt at lower temperatures? I mean, the performance of Ni alloys above their melting temperatures is obviously poor, even without comparing “virtual” and actual values.

Author's response to the comment:

To compare the values of diffusion coefficients in different solids, a homological temperature scale T / Tm is usually used, and this method provides useful information in the field of "diffusion physics".  However, to compare the structural stability of different alloys at high temperatures, we recommend using our proposed method.

Comment 9:

Row 389 is unclear to me, I don’t understand “δ = 9.7%, Ϭ0.2 = 1262 MPa.” I guess they are elongation to fracture and 0.2% proof stress. Please use the subscript also in this case. (similar comment on row 396)

Author's response to the comment:

The results of testing the mechanical properties of alloys are taken by us from the articles, links to which are given. Of course, Ϭ0.2 = σ0.2 = 1262 MPa.

Comment 10:

In section 3.2 you include Ti among refractory metals, but its melting temperature is about 1600°C, is it correct? I don’t understand why you mention (row 391) the use of powder metallurgy to produce this alloys, in my opinion this information looks somewhat uncorrelated to the rest of the paragraph. At row 396 there is again a symbol which is not defined, σB.

Author's response to the comment:

A set of refractory metals (in ascending melting temperatures) begins with Ti (see, for example, Refractory metals in mechanical engineering. Guide. Edited by A.T. Tumanov and K.I. Portnoy. M., Mechanical Engineering, 1967. – 392). Among the technologies to produce alloys, we attract the most attention to powder metallurgy in the hope that these technologies will be used for the production of new alloys.

Comment 11:

At row 412 you use “plasticity” but I guess that you are referring to “ductility”. I guess this happens a few times throughout the paper, so please verify that the term is appropriate.

Author's response to the comment:

We have replaced the word “plasticity” with “ductility”.

Comment 12:

Check the symbols in 421-423 (what is ψ?)

Author's response to the comment:

We have excluded the symbol ψ from the article.

That Note: corrections made are highlighted in yellow in the manuscript.

Reviewer 3 Report

lines 198-206: This paragraph is doubled here, please correct.

l389, l422 '60.2': Please correct symbols in the manuscript. Here e.g. to sigma_{0.2}.

Author Response

Comments and Suggestions for Authors

Comment 1:

 - lines 198-206: This paragraph is doubled here, please correct.

Author's response to the comment:

Thanks. We fixed it.

Comment 2:

 - l389, l422 '60.2': Please correct symbols in the manuscript. Here e.g. to sigma_{0.2}.

Author's response to the comment:

We tried to correct all the symbols – we hope we succeeded.

That Note: corrections made are highlighted in yellow in the manuscript.

Round 2

Reviewer 1 Report

The authors have adequately addressed the comments. Now the manuscript is ready for publicaton.

Reviewer 3 Report

The work can be accepted in the present form.